# Neurosymbolic Association Rule Mining from Tabular Data

**Erkan Karabulut**[*] **Paul Groth Victoria Degeler**

*University of Amsterdam*

**Editors:** Leilani H. Gilpin, Eleonora Giunchiglia, Pascal Hitzler, and Emile van Krieken

## Abstract

Association Rule Mining (ARM) is the task of mining patterns among data features in the form of logical rules, with applications across a myriad of domains. However, high-dimensional datasets often result in an excessive number of rules, increasing execution time and negatively impacting downstream task performance. Managing this rule explosion remains a central challenge in ARM research. To address this, we introduce Aerial+, a novel neurosymbolic ARM method. Aerial+ leverages an under-complete autoencoder to create a neural representation of the data, capturing associations between features. It extracts rules from this neural representation by exploiting the model's reconstruction mechanism. Extensive evaluations on five datasets against seven baselines demonstrate that Aerial+ achieves state-of-the-art results by learning more concise, high-quality rule sets with full data coverage. When integrated into rule-based interpretable machine learning models, Aerial+ significantly reduces execution time while maintaining or improving accuracy.

## 1. Introduction

Association Rule Mining (ARM) is a knowledge discovery task that aims to *mine* commonalities among features of a given dataset as logical implications (Agrawal et al., 1994). It has a plethora of applications in various domains including healthcare (Zhou et al., 2020), recommendation systems (Roy and Dutta, 2022), and anomaly detection (Sarno et al., 2020). Beyond knowledge discovery, ARM plays a crucial role in rule-based interpretable Machine Learning (ML) models such as rule list classifiers (Angelino et al., 2018), particularly in high-stakes decision-making (Rudin, 2019). Such models construct interpretable predictive models using pre-mined rules from ARM algorithms and class labels (Liu et al., 1998; Letham et al., 2015; Angelino et al., 2018). In this paper, we focus on ARM applied to tabular data, a common area of study in ARM research (Luna et al., 2019).

The high dimensionality of data in state-of-the-art ARM methods leads to the generation of an excessive number of rules and prolonged execution times. This remains a significant research problem in the ARM literature (Telikani et al., 2020; Kaushik et al., 2023). This problem propagates to downstream tasks in rule-based models as processing a high number of rules is resource-intensive. The most popular solutions to this problem include constraining data features (i.e. ARM with item constraints (Srikant et al., 1997; Baralis et al., 2012; Yin et al., 2022)) and mining top-k high-quality rules based on a rule quality criteria (Fournier-Viger et al., 2012; Nguyen et al., 2018).

To address this research problem, we make the following contributions: i) a novel neurosymbolic ARM method - *Aerial+* (Section 3); ii) two comprehensive evaluations of Aerial+

---

[*] Corresponding author: e.karabulut@uva.nl

(Section 4) on 5 real-world tabular datasets (Kelly et al., 2023) demonstrating Aerial+'s superiority over seven baselines in knowledge discovery and downstream classification tasks.

Aerial+ consists of two main steps: the creation of a neural representation of the data using an under-complete denoising autoencoder (Vincent et al., 2008) which captures the associations between features, and the extraction of rules from the neural representation by exploiting the reconstruction mechanism of the autoencoder. The first evaluation uses rule quality, the standard method in ARM literature (Luna et al., 2019; Kaushik et al., 2023), which shows that Aerial+ can learn a more concise set of high-quality rules than the state-of-the-art with full data coverage. While prior work on ARM predominantly evaluates rule quality, we further evaluate Aerial+ on downstream classification tasks, as part of popular rule-based interpretable ML models such as CORELS (Angelino et al., 2018). The results show that the smaller number of rules learned by Aerial+ leads to faster execution times with similar or higher accuracy. These findings indicate that Aerial+, a neurosymbolic approach, can effectively address the rule explosion problem in ARM research.

## 2. Related Work

This paper focuses on ARM applied to tabular data. We first give the original definition of ARM following from (Agrawal et al., 1994) and then discuss current ARM methods.

**Association rules.** Let $I = \{i_1, i_2, ..., i_m\}$ be a full set of items. Let $X, Y \subseteq I$ be subsets of $I$, called *itemsets*. An *association rule*, denoted as $X \rightarrow Y$ (*'if X then Y'*), is a first-order horn clause which has at most one positive literal ($|Y| = 1$) in its Conjunctive Normal Form (CNF) form ($\neg X \vee Y$), and $X \cap Y = \varnothing$. The itemset $X$ is referred to as the *antecedent*, while $Y$ is the *consequent*. Let $D = \{T_1, T_2, ..., T_n\}$ be a set of transactions where $\forall T \in D$, $T \subseteq I$, meaning that each transaction consists of a subset of items in I. An association rule $X \rightarrow Y$ is said to have *support* $s \in [0, 1]$ if a fraction $s$ of the transactions in the dataset $D$ contain the itemset $X \cup Y$. The *confidence* of a rule is the conditional probability that a transaction containing $X$ also contains $Y$. ARM is a knowledge discovery task that aims to find association rules that satisfy predefined support, confidence, or other rule quality metrics in a given transaction set. In practice, tabular data is usually transformed into transaction format using (one-hot) encoding to enable ARM.

**Rule explosion.** ARM has a number of algorithms for finding exhaustive solutions, such as FP-Growth (Han et al., 2000), and HMine (Pei et al., 2007). An extensive survey is provided by Luna et al. (2019). However, ARM suffers from high data dimensionality, leading to excessive rules and long execution times (Telikani et al., 2020; Kaushik et al., 2023). A common remedy is to run ARM with item constraints (Srikant et al., 1997) that focuses on mining rules for the items of interest rather than all (Baralis et al., 2012; Shabtay et al., 2021). Closed itemset mining (Zaki and Hsiao, 2002) further reduces rule redundancy by identifying only frequent itemsets without frequent supersets of equal support. Another solution is to mine top-k rules based on a given rule quality criteria aiming to control the number of rules to be mined (Fournier-Viger et al., 2012). These methods optimize ARM by reducing search space and improving execution times by limiting rule generation. Aerial+ is orthogonal to these methods and can be fully integrated with them.

**Numerical ARM.** Another aspect of ARM is its application to numerical data, where many approaches leverage nature-inspired optimization algorithms. Kaushik et al. (2023) presents a comprehensive survey of such algorithms. Numerical ARM methods aim to find

feature intervals that produce high-quality rules based on predefined fitness functions, combining rule quality criteria. The best performing methods include Bat Algorithm (Yang, 2010; Heraguemi et al., 2015), Grey Wolf Optimizer (Yildirim and Alatas, 2021; Mirjalili et al., 2014), Sine Cosine Algorithm (Mirjalili, 2016; Altay and Alatas, 2021), and Fish School Search (Bastos Filho et al., 2008; Bharathi and Krishnakumari, 2014). Fister et al. (2018) extends numerical ARM methods to categorical data. While numerical ARM methods do not primarily aim to address the rule explosion problem, we include these methods in our evaluation for completeness, as they may yield fewer rules than exhaustive approaches.

**Interpretable ML.** Besides knowledge discovery, ARM is widely used in rule-based interpretable ML models, which is the standard approach to high-stake decision-making (Rudin, 2019). Examples include associative classifiers such as CBA (Liu et al., 1998), rule set, and rule list learners (Letham et al., 2015; Angelino et al., 2018) that construct rule-based classifiers from pre-mined rules or frequent itemsets via ARM and class labels. Since these models rely on ARM to pre-mine rules, the excessive number of rules and long execution times carry over to downstream interpretable ML tasks, further increasing computational costs. All these methods work with exhaustive ARM approaches such as FP-Growth to pre-mine frequent itemsets and rules. Numerous versions of FP-Growth have also been proposed to alleviate the aforementioned issues such as Guided FP-Growth (Shabtay et al., 2021) for ARM with item constraints, parallel FP-Growth (Li et al., 2008) and FP-Growth on GPU (Jiang and Meng, 2017) for better execution times.

**Deep Learning (DL) in ARM.** To the best of our knowledge, very few DL-based methods can directly mine association rules from tabular data, despite DL's widespread success. Patel and Yadav (2022) used an autoencoder (Bank et al., 2023) to mine frequent itemsets from a grocery dataset and derive association rules, but their study lacks an explicit algorithm or source code. Berteloot et al. (2024) introduced ARM-AE, another autoencoder-based ARM method. ARM-AE was not extensively evaluated and yields low-confidence rules as reported in their paper (e.g., 33% confidence in one dataset) and our findings (Section 4.1). Karabulut et al. (2024) proposed a DL-based ARM leveraging autoencoders, however, it is tailored to Internet of Things (IoT) domain, incorporating sensor data and knowledge graphs. Note that the term *rule learning* encompasses different tasks, such as learning rules over graphs (Ho et al., 2018), which are out of scope for this paper.

**Proposed solution.** To address the challenges of rule explosion and high execution time, we turn towards a neurosymbolic approach that uses DL to handle high dimensionality. The aim is to complement existing methods such as (i) ARM with item constraints, and (ii) top-k rule mining. Additionally, to further address execution time, parallel execution on GPUs should be supported.

## 3. Methodology

This section presents our neurosymbolic ARM method *Aerial+* for tabular data.

### 3.1. Neurosymbolic Rule Mining Pipeline

Figure 1 illustrates the pipeline of operations for Aerial+. Following the ARM literature, we convert tabular data into transactions by applying one-hot encoding (e.g., Berteloot et al. (2024)). Each transaction is taken as a vector and fed into an under-complete denoising autoencoder (Vincent et al., 2008) to create a neural representation of the data. An under-

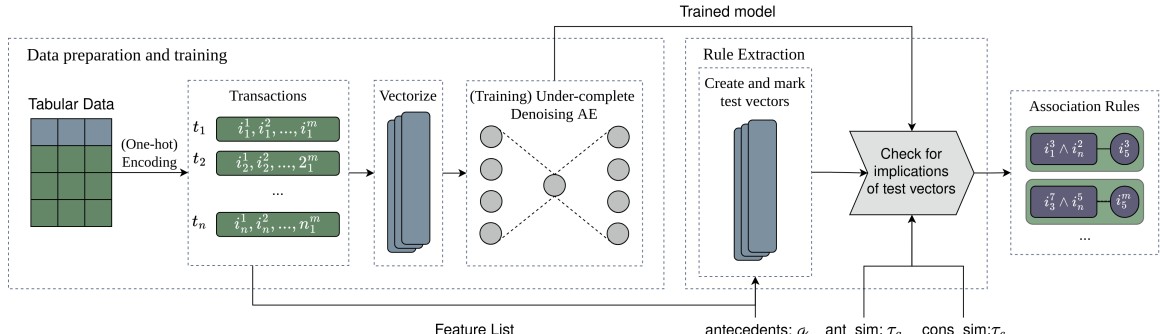

Figure 1: Neurosymbolic ARM pipeline of Aerial+ for tabular data.

complete autoencoder creates a lower-dimensional representation of the data, encoding its prominent features. The denoising mechanism makes the model robust to noise. The model is trained to output a probability distribution per feature, ensuring category probabilities add up to 1 (Section 3.2). After training, the model enters the rule extraction stage, where *test vectors* are created, each matching the input feature dimensions. Categories of interest, say $X$, are marked in test vectors by assigning a probability of 1 (100%). A forward pass through the trained model is performed with each of the test vectors, and if the output probability for a set of feature categories $Y$ exceeds a threshold, the marked categories $X$ are said to imply $Y$, forming association rules $X \to Y$ (Section 3.3).

### 3.2. Autoencoder Architecture and Training Stage

Let $F = \{f_1, f_2, ..., f_k\}$ be a set of $k$ features in a tabular dataset (e.g. columns in a table) and $f_i^{1...c_i}$ represent categories (possible values) for feature $f_i$ ($1 \leq i \leq k$) where $c_i = |f_i|$ indicates the number of categories for feature $f_i$. To represent such a tabular dataset as an ARM problem as defined in Section 2, we define the full set of items $I$ as consisting of all possible categories of all features $I = \{f_1^1, ..., f_1^{c_1}, ..., f_k^1, ..., f_k^{c_k}\}$. Each transaction $T$ in the tabular dataset (e.g., a row in a table) is represented as an itemset $T \subset I$, where each feature contributes exactly one item: $\forall i \in \{1, ..., k\}, \exists! j \in \{1, ..., c_i\}$ such that $f_i^j \in T$. Using one-hot encoded representation, $f_i^{1...c_i} = \{0, 1\}$, with 0 and 1 indicating the absence or the presence of a feature category in a given transaction.

The input to the autoencoder consists of vectors of dimension $\sum_{i=1}^{k} c_i$. Next, a random noise $N \sim [-0.5, 0.5]$ is added to each feature category $f_i^j$ ($1 \leq j \leq c_i$), with values clipped to $[0, 1]$ using $f_i^j = \min(1, \max(0, f_i^j + N))$.

This noisy input is passed through an autoencoder with decreasing dimensions (each layer has half the parameters of the previous) and one to three layers per encoder and decoder. The number of layers, training epochs, and batch sizes depend on dataset dimensions and instance counts. More than three layers or over two training epochs did not improve performance. $tanh(z) = \frac{e^z - e^{-z}}{e^z + e^{-z}}$ is used as the activation function in hidden layers. After encoding and decoding, the softmax function $\sigma$ is applied per feature, $\sigma(z_i) = \frac{e^{z_i}}{\sum_{j=1}^{n} e^{z_j}}$, ensuring values for each feature's categories sum to 1 (100%):

$$\sum_{j=1}^{c_i} \sigma(f_i^j) = \sum_{j=1}^{c_i} \frac{e^{f_i^j}}{\sum_{k=1}^{c_i} e^{f_i^k}} = \frac{\sum_{j=1}^{c_i} e^{f_i^j}}{\sum_{k=1}^{c_i} e^{f_i^k}} = 1$$

---

**Algorithm 1:** Aerial+'s rule extraction algorithm from a trained autoencoder

---

**Input:** Trained autoencoder: $AE$, max antecedents: $a$, similarity thresholds $\tau_a, \tau_c$
**Output:** Extracted rules $\mathcal{R}$

1   $\mathcal{R} \leftarrow \emptyset$, $\mathcal{F} \leftarrow AE.input\_feature\_categories$;
2   **for** $i \leftarrow 1$ **to** $a$ **do**
3      $\mathcal{C} \leftarrow \binom{\mathcal{F}}{i}$;
4      **foreach** $S \in \mathcal{C}$ **do**
5          $\mathbf{v}_0 \leftarrow$ UniformProbabilityVectorPerFeature($\mathcal{F}$);
6          $\mathcal{V} \leftarrow$ MarkFeatures($S, \mathbf{v}_0$);
7          **foreach** $\mathbf{v} \in \mathcal{V}$ **do**
8              $\mathbf{p} \leftarrow AE(\mathbf{v})$;
9              **if** $\min\limits_{f \in S} p_f < \tau_a$ **then**
10                  $S$.low_support $\leftarrow$ **True**;
11                  **continue** with the next $\mathbf{v}$;
12              **foreach** $f \in \mathcal{F} \setminus S$ **do**
13                  **if** $p_f > \tau_c$ **then** $\mathcal{R} \leftarrow \mathcal{R} \cup \{(S \rightarrow f)\}$
14      $\mathcal{F} \leftarrow \{f \in \mathcal{F} \mid f.$low_support $=$ **False**$\}$;
15   **return** $\mathcal{R}$;

---

Binary cross-entropy (BCE) loss is applied per feature, and the results are aggregated:

$$BCE(F) = \sum_{i=1}^{k} BCE(f_i) = \sum_{i=1}^{k} \frac{1}{c_i} \sum_{j=1}^{c_i} -\left(y_{i,j} \log(p_{i,j}) + (1 - y_{i,j}) \log(1 - p_{i,j})\right)$$

where $p_{i,j}$ refers to $\sigma(f_i^j)$ and $y_{i,j}$ refers to initial noise-free version of $f_i^j$. Finally, the learning rate is set to $5E - 3$. The Adam (Kingma and Ba, 2014) optimizer is used for gradient optimization with a weight decay of $2E - 8$.

### 3.3. Rule Extraction Stage

This section describes Aerial+'s rule extraction process from a trained autoencoder.

**Intuition.** Autoencoders can learn a neural representation of data, and this representation includes the associations between the features. We hypothesize that the reconstruction ability of autoencoders can be used to extract associations. After training, if a forward run on the trained model with a set of marked categories $A$ results in

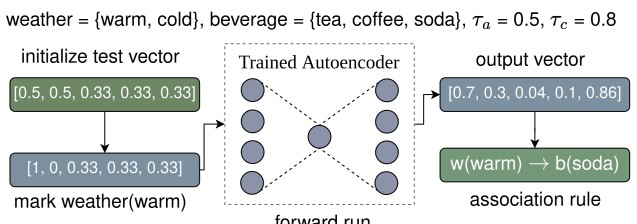

Figure 2: Aerial+ rule extraction example.

successful reconstruction (high probability) of categories $C$, we say that marked features $A$ imply the successfully reconstructed features $C$, such that $A \rightarrow C \setminus A$ (no self-implication).

**Example.** Figure 2 shows rule extraction. Let *weather* and *beverage* be features with categories cold, warm and tea, coffee, soda, respectively. We start with a test vector of size 5, assigning equal probabilities per feature: $[0.5, 0.5, 0.33, 0.33, 0.33]$. Then we mark

$weather(warm)$ by assigning 1 to $warm$ and 0 to $cold$, $[1, 0, 0.33, 0.33, 0.33]$, and call the resulting vector a *test vector*. Assume that after a forward run, $[0.7, 0.3, 0.04, 0.1, 0.86]$ is received as the output probabilities. Since the probability of $p_{weather(warm)} = 0.7$ is bigger than the given antecedent similarity threshold ($\tau_a = 0.5$), and $p_{beverage(soda)} = 0.86$ probability is higher than the consequent similarity threshold ($\tau_c = 0.8$), we conclude with $weather(warm) \rightarrow beverage(soda)$.

**Algorithm.** The rule extraction process is given in Algorithm 1. Line 1 stores input feature categories into $\mathcal{F}$. Lines 2-14 iterate over the number of antecedents $i$ and line 3 generates an $i$-feature combination $\mathcal{F}$. Lines 4-13 iterate over the feature combinations $\mathcal{C}$. For each combination $S$, line 5 creates a vector with uniform probabilities per feature category, e.g., $[0.5, 0.5, 0.33, 0.33, 0.33]$ vector in the example given above. Line 6 creates a set of test vectors where a combination of feature categories in $S$ are marked per test vector. This corresponds to the $[1, 0, 0.33, 0.33, 0.33]$ vector in the example where $weather(warm)$ was marked. Lines 7-13 iterate over the test vectors. Line 8 performs a forward run on the trained autoencoder $AE$ with each test vector $v$. Lines 9-11 compare the output probabilities corresponding to $S$ with a given antecedent similarity threshold $\tau_a$, and the algorithm continues with high probability $S$ values. Lines 12-13 compare the output probabilities for the categories in $\mathcal{F}$ that are not in $S$ already ($F \setminus S$), with a given consequent similarity threshold $\tau_c$ and stores categories with high probability. Finally, line 14 removes the low support categories from $\mathcal{F}$ so they are ignored in line 3 of the next iteration.

Appendix B describes **hyperparameter** tuning process for the thresholds $\tau_a$ and $\tau_c$. A **formal justification** for Aerial+'s use in ARM is given in Appendix F.

**Scalability.** A **runtime complexity** analysis (Appendix A) shows that Aerial+ scales linearly with the number of transactions ($n$) during training and polynomially with the number of features ($k$) during rule extraction. Besides Algorithm 1, two Aerial+ variants for ARM with item constraints and frequent itemset mining are given in Appendix C, along with explanations of more variants. Since each feature combination $S \in \mathcal{C}$ is processed independently, Algorithm 1 is parallelizable. All computations use vectorized operations, enabling efficient GPU acceleration. Extrapolating the execution times in Section 4, Aerial+ can scale to tens of thousands of features on a laptop (see **Hardware** below) in a day.

## 4. Evaluation

Two sets of experiments evaluate Aerial+ thoroughly: (1) rule quality assessment (Section 4.1), a standard method in ARM research (Luna et al., 2019), and (2) testing on downstream classification tasks (Section 4.2) providing input rules to interpretable rule-based classifiers commonly used in high-stake decision-making (Rudin, 2019).

**Hardware.** All experiments were run on a 12th Gen Intel® Core™ i5-1240P × 16 CPU, with 16 GiB memory, and 512 GB disk. No GPUs were used, and no parallel execution was conducted.

**Open-source.** The source code of Aerial+, all the baselines and datasets are open-source and

Table 1: Datasets from the UCI ML repository.

| Dataset | Features | Feature Categories | Instances |
|---|---|---|---|
| Breast Cancer | 9 | 43 | 286 |
| Congressional Voting Records | 16 | 48 | 435 |
| Mushroom | 22 | 117 | 8124 |
| Chess (King-Rook vs. King-Pawn) | 35 | 71 | 3196 |
| Spambase | 57 | 155 | 4601 |

can be found at: https://github.com/DiTEC-project/aerial-rule-mining, and Aerial+'s **Python package** at: https://github.com/DiTEC-project/pyaerial.

**Datasets.** The experiments use five UCI ML (Kelly et al., 2023) datasets (Table 1), a standard ARM benchmark. Numerical features are discretized into 10 intervals via equal-frequency binning (Foorthuis, 2020) for algorithms that require it.

### 4.1. Experimental Setting 1: Execution Time and Rule Quality Evaluation

The goal of this experimental setting is to compare Aerial+ with seven state-of-the-art ARM algorithms which are given in Table 2 together with their parameters. The comparison is based on the **standard evaluation criteria** in ARM literature: execution time, number of rules, average support and confidence per rule, and data coverage for the whole dataset.

**Challenges in comparison.** Comparing different algorithm types is inherently challenging due to their distinct characteristics. Exhaustive methods identify all rules meeting a given support and confidence threshold, while optimization-based approaches operate within a predefined evaluation limit, improving results up to a point. DL-based ARM methods depend on similarity thresholds for rule quality. Given these differences, we made our best effort to compare algorithms fairly and showed the **trade-offs under different conditions**. Optimization-based methods were implemented using NiaARM (Stupan and Fister, 2022) and NiaPy (Vrbančič et al., 2018), with original parameters, while exhaustive methods used Mlxtend (Raschka, 2018). Antecedent length in exhaustive and DL-based methods is fixed at 2 unless stated otherwise (not controllable in others). The minimum support threshold for exhaustive methods is set to half the average support of Aerial+ rules for comparable support values, and ARM-AE's number of rules per consequent (N) is set to Aerial+'s rule count divided by the number of categories to ensure comparable rule counts.

**Execution time and number of rules.** Figure 3 shows changes in rule count (bars, left y-axis) and execution time (lines, right y-axis) for exhaustive methods as the number of antecedents increases (top, min_support=0.05) or minimum support decreases (bottom, antecedents=2). Results show that exhaustive methods produce a substantially higher number of rules and has longer execution times with more antecedents or lower support thresholds. Rule counts reach millions for relatively larger datasets (Chess, Spambase) after 3–4 antecedents, while execution time reaches hours.

Table 3 shows that optimization-based ARM requires long evaluations, hence execution times, to yield higher quality rules. However, improvement in rule quality stagnates after 50,000 evaluations. The results are consistent across datasets (see Appendix D). In contrast,

Table 2: Aerial+ and baselines with their parameters (R = Aerial+ rules, C = categories).

| Algorithm | Type | Parameters |
|---|---|---|
| Aerial+ | DL-based | a = 2, $\tau_a = 0.5, \tau_c = 0.8$ |
| ARM-AE | DL-based | M=2, N=$|R|/|C|$, L=0.5 |
| Bat Algorithm (BAT) | Optimization | initial_population=200, |
| Grey Wolf Optimizer (GWO) | Optimization | max_evaluations=50000, |
| Sine Cosine Algorithm (SC) | Optimization | optimization_objective=(support, |
| Fish School Search (FSS) | Optimization | confidence) |
| FP-Growth | Exhaustive | antecedents = 2, |
| HMine | Exhaustive | min_support=$\frac{1}{2}\mathbb{E}$[support($R$)], min_conf=0.8 |

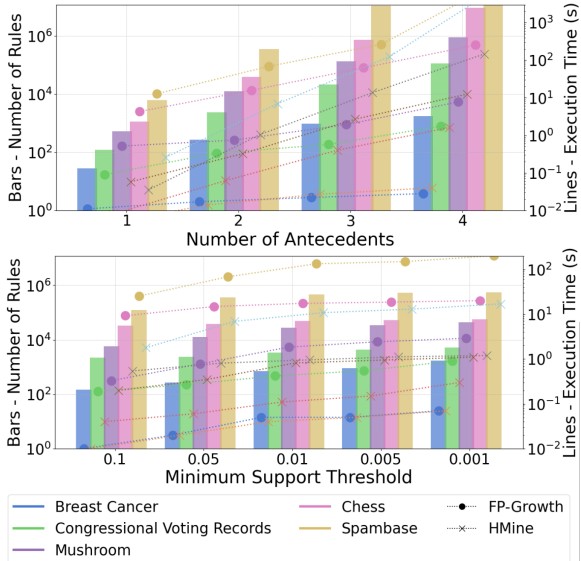

Figure 3: Exhaustive methods incur higher execution times as antecedents increase (top) or support threshold decreases (bottom).

| Evals. | Algorithm | # Rules | Time (s) | Conf. |
|---|---|---|---|---|
| 1000 | BAT | 14.3 | 2.42 | 0.52 |
| | GWO | 39.3 | 3.59 | 0.31 |
| | SC | 0.5 | 2.99 | 0.12 |
| | FSS | 2.3 | 3.7 | 0.5 |
| 10000 | BAT | 1233 | 38.02 | 0.62 |
| | GWO | 490.9 | 52.44 | 0.54 |
| | SC | 0.6 | 67.22 | 0.13 |
| | FSS | 25.4 | 71.89 | 0.27 |
| 50000 | BAT | 1377.2 | 225.57 | 0.62 |
| | GWO | 1924.1 | 184.56 | 0.63 |
| | SC | 1.33 | 281.84 | 0.48 |
| | FSS | 794.9 | 352.99 | 0.38 |
| 100000 | BAT | 1335.4 | 295.95 | 0.62 |
| | GWO | 3571.6 | 300.64 | 0.57 |
| | SC | 1 | 432.17 | 0.2 |
| | FSS | 6830.8 | 536.59 | 0.49 |

Table 3: Optimization-based methods need long evaluations for good performance (Mushroom). The results are consistent across datasets (Appendix D).

Figure 4 shows that as the number of antecedents increases, Aerial+ generates fewer rules and achieves lower execution times than exhaustive methods (Figure 3). It also significantly outperforms optimization-based methods (Table 3), even with 4 antecedents compared to running the optimization-based method for 10,000 evaluations or more. Note that the execution time for Aerial+ includes **both** training and rule extraction.

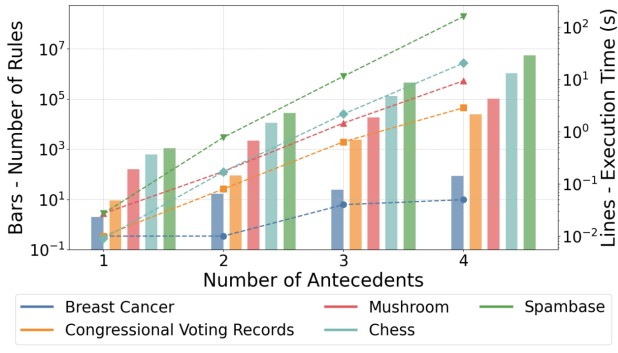

Figure 4: Aerial+ yields fewer rules and lower execution time as antecedents increase.

**Rule Quality.** Table 4 presents rule quality experiment results. **How to interpret the results?** Since there is **no single criterion** to evaluate rule quality in ARM literature, we take *having a concise set of high-confidence rules with full data coverage in a practical duration* as the main criterion. The results show that Aerial+ has the most concise number of rules with full data coverage and higher or compatible confidence levels to the exhaustive methods on all datasets. Aerial+ runs significantly faster than the exhaustive methods on larger datasets. Exhaustive methods resulted in high-confidence rules as specified by its parameters, however, 2 to 10 times higher number of rules than Aerial+.

ARM-AE produced the lowest confidence levels with high execution times on relatively larger datasets (Chess and Spambase). Optimization-based methods led to the second lowest confidence rules on average with the highest execution time. On the Spambase dataset, the optimization-based methods could not find rules despite running them significantly longer than others. Note that, as given in Table 6 in Appendix D, running optimization-based methods even longer allowed them to find rules, however, with low confidence.

Table 4: Aerial+ can find a more concise set of high-quality rules with full data coverage and runs faster on large datasets (Cov=Coverage, Conf=Confidence, FP-G=FP-Growth).

| Algorithm | #Rules | Time (s) | Cov. | Support | Conf. | Algorithm | #Rules | Time (s) | Cov. | Support | Conf. |
|---|---|---|---|---|---|---|---|---|---|---|---|
| **Congressional Voting Records** | | | | | | **Breast Cancer** | | | | | |
| BAT | 1913 | 208 | 1 | 0.06 | 0.45 | BAT | 787.1 | 162.18 | 1 | 0.07 | 0.41 |
| GW | 2542 | 186 | 1 | 0.05 | 0.48 | GW | 1584 | 129.18 | 1 | 0.08 | 0.42 |
| SC | 7 | 186 | 0.46 | 0.01 | 0.43 | SC | 33.6 | 137.66 | 1 | 0.03 | 0.27 |
| FSS | 10087 | 272 | 1 | 0.01 | 0.71 | FSS | 6451.6 | 225.71 | 1 | 0.02 | 0.36 |
| FP-G \| HMine | 1764 | 0.09 \| 0.04 | 1 | 0.29 | 0.88 | FP-G \| HMine | 94 | 0.01 \| 0.01 | 1 | 0.34 | **0.87** |
| ARM-AE | 347 | 0.21 | 0.03 | 0.23 | 0.45 | ARM-AE | 131 | 0.09 | 0.01 | 0.19 | 0.27 |
| **Aerial+** | 149 | 0.25 | 1 | 0.32 | **0.95** | **Aerial+** | 50 | 0.19 | 1 | 0.39 | 0.86 |
| **Mushroom** | | | | | | **Chess** | | | | | |
| BAT | 1377.2 | 225.57 | 1 | 0.1 | 0.62 | BAT | 2905.9 | 235.34 | 1 | 0.17 | 0.64 |
| GW | 1924.1 | 184.56 | 1 | 0.11 | 0.63 | GW | 5605.25 | 255.56 | 1 | 0.31 | 0.65 |
| SC | 1.33 | 281.84 | 0.07 | 0.02 | 0.48 | SC | 1 | 545.71 | 0 | 0 | 0.7 |
| FSS | 794.9 | 352.99 | 1 | 0.04 | 0.38 | FSS | 32.75 | 380.73 | 0.4 | 0 | 0.36 |
| FP-G \| HMine | 1180 | 0.1 \| 0.07 | 1 | 0.43 | 0.95 | FP-G \| HMine | 30087 | 12.43 \| 0.7 | 1 | 0.46 | 0.93 |
| ARM-AE | 390 | 0.33 | 0 | 0.22 | 0.23 | ARM-AE | 22052 | 26.98 | 0.02 | 0.39 | 0.54 |
| **Aerial+** | 321 | 0.38 | 1 | 0.44 | **0.96** | **Aerial+** | 16522 | 0.22 | 1 | 0.45 | **0.95** |
| **Spambase** | | | | | | | | | | | |
| BAT | 0 | 424 | No rules found | | | | | | | | |
| GW | 0 | 508 | No rules found | | | | | | | | |
| SC | 0 | 643 | No rules found | | | | | | | | |
| FSS | 0 | 677 | No rules found | | | | | | | | |
| FP-G \| HMine | 125223 | 21.4 \| 2.14 | 1 | 0.64 | 0.92 | | | | | | |
| ARM-AE | 85327 | 254 | 0.03 | 0.31 | 0.38 | | | | | | |
| **Aerial+** | 43996 | 1.92 | 1 | 0.62 | **0.97** | | | | | | |

## 4.2. Experimental Setting 2: Aerial+ on Downstream Tasks

This setting evaluates Aerial+ on rule-based classification tasks.

**Setup.** CBA (M2) (Liu et al., 1998), Bayesian Rule List learner (BRL) (Letham et al., 2015) and Certifiably Optimal RulE ListS (CORELS) (Angelino et al., 2018) are well-known rule-based classifiers that work with either pre-mined association rules (CBA) or frequent itemsets (BRL and CORELS).[1] Rule-based classifiers perform pre-mining using exhaustive methods such as FP-Growth with low minimum support thresholds to ensure a wide pool of options when building the classifiers. Given this and exhaustive methods having the second highest rule quality after Aerial+ in Experimental Setting 1, we run the rule-based classifiers with an exhaustive method (FP-Growth) and Aerial+ with 2 antecedents for comparison.

FP-Growth is run with a 1% min support threshold (and 80% min. confidence for CBA, as confidence applies only to rules and not frequent itemsets) for CBA and CORELS. For BRL, we use a 10% minimum support threshold to avoid impractical execution times on our hardware with lower thresholds. Note that depending on dataset features, different support thresholds may yield different outcomes, which are analyzed in Appendix E due to space constraints. The learned rules or frequent itemsets are passed to the classifiers for classification, followed by **10-fold cross-validation**.[2]

Table 5 shows the experimental results including the number of rules (CBA) or frequent itemsets (BRL and CORELS), accuracy, and execution times.[3] The results show that with a significantly smaller number of rules (or frequent itemsets) with Aerial+, rule-based classifiers run substantially faster than with the rules from FP-Growth. Despite having a

---

1. We created a version of Aerial+ for frequent itemset mining in Appendix C to run BRL and CORELS.

2. CBA uses pyARC (Filip and Kliegr, 2018), while BRL and CORELS use imodels (Singh et al., 2021).

3. The execution time includes rule mining (including training for Aerial+) and classifier construction time.

Table 5: Rule-based interpretable ML models with Aerial+ achieve higher or comparable accuracy with significantly lower execution time. Bold indicates the highest performance.

| Dataset | Algorithm | # Rules or Items | Accuracy | Exec. Time (s) |
|---|---|---|---|---|
| | | Exhaustive \| Aerial+ | Exhaustive \| Aerial+ | Exhaustive \| Aerial+ |
| Congressional Voting Records | CBA | 3437 \| **1495** | 91.91 \| **92.66** | 0.34 \| **0.14** |
| | BRL | 2547 \| **57** | **96.97** \| **96.97** | 15.37 \| **9.69** |
| | CORELS | 4553 \| **61** | **96.97** \| **96.97** | 3.04 \| **0.17** |
| Mushroom | CBA | 27800 \| **2785** | **99.82** \| **99.82** | 1.75 \| **1.30** |
| | BRL | 5093 \| **493** | **99.87** \| 99.82 | 244 \| **167** |
| | CORELS | 23271 \| **335** | 90.14 \| **99.04** | 61 \| **2** |
| Breast Cancer | CBA | 695 \| **601** | 66.42 \| **71.13** | **0.08** \| 0.28 |
| | BRL | 2047 \| **290** | 71.13 \| **71.46** | 16.82 \| **14.5** |
| | CORELS | 2047 \| **369** | 73.69 \| **75.82** | 1.42 \| **0.40** |
| Chess | CBA | 49775 \| **34490** | **94.02** \| 93.86 | 24.31 \| **6.24** |
| | BRL | 19312 \| **1518** | **96.21** \| 95.93 | 321 \| **119** |
| | CORELS | 37104 \| **837** | 81.1 \| **93.71** | 106 \| **3.87** |
| Spambase | CBA | 125223 \| **33418** | 84.5 \| **85.42** | 23.87 \| **7.56** |
| | BRL | 37626 \| **5190** | 72.78 \| **84.93** | 1169 \| **431** |
| | CORELS | 275003 \| **1409** | 85.37 \| **87.28** | 1258 \| **5.23** |

significantly lower number of rules, all of the rule-based classifiers with rules (or frequent itemsets) from Aerial+ lead to a higher or comparable accuracy on all datasets.

**Hyperparameter analysis.** Appendix B (omitted here for brevity) shows how similarity thresholds ($\tau_a$, $\tau_c$) affect Aerial+ rule quality.

**Qualitative advantages of Aerial+.** Aerial+ offers two qualitative advantages over the state-of-the-art: i) once trained in O(n) time, it can *verify* whether a given association exists by creating the corresponding *test vector* in O(1) time—unlike other methods, ii) it can be integrated into larger neural networks for interpretability.

## 5. Conclusions

This paper introduced Aerial+, a novel neurosymbolic association rule mining method for tabular data, to address rule explosion and high execution time challenges in ARM research. Aerial+ uses an under-complete autoencoder to create a neural representation of the data and extracts association rules by exploiting the model's reconstruction mechanism.

Extensive rule quality evaluations in Section 4.1 show that Aerial+ learns a compact set of high-quality association rules with full data coverage, outperforming state-of-the-art methods in execution time on high-dimensional datasets. In downstream classification tasks within rule-based interpretable models (Section 4.2), Aerial+'s concise rules significantly reduce execution time while maintaining or improving accuracy. Aerial+ supports parallel and GPU execution (Section 3.3), and results across rule quality, downstream tasks, and runtime complexity demonstrate its scalability on large datasets. Additionally, two Aerial+ variants support ARM with item constraints and frequent itemset mining. We also describe how other solutions to rule explosion can be integrated into Aerial+.

Overall, our empirical findings show that combining deep learning's capability to handle high-dimensional data with algorithmic solutions, as in Aerial+, to do rule mining can address longstanding problems in ARM research. Future work will explore the potential of other deep learning architectures for learning associations.

**Acknowledgement.** This work has received support from the Dutch Research Council (NWO), in the scope of the Digital Twin for Evolutionary Changes in water networks (DiTEC) project, file number 19454.

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

## Appendix A. Runtime Complexity Analysis of Aerial+

This section provides a step-by-step runtime complexity analysis of the proposed Algorithm 1 in big O notation.

Line 1 initializes $\mathcal{R}$ and $\mathcal{F}$ in O(1) time.

Lines 2-14 iterate over the number of antecedents $a$ for the outer loop, meaning the outer loop runs $O(a)$ times.

Line 3 calculates i-feature combinations over $\mathcal{F}$, denoted as $\mathcal{C}$. The number of such subsets are $\binom{|\mathcal{F}|}{i}$ ($1 \leq i \leq a$), hence $O\binom{|\mathcal{F}|}{i}$ which is re-written as $O(|\mathcal{F}|^i)$.

Lines 4-13 iterate over the feature subsets $\mathcal{S} \in \mathcal{C}$ and for each subset:

1. Creates a uniform probability vector of size $|\mathcal{F}|$ in $O(|\mathcal{F}|)$ time.

2. Creates test vectors of the same size as the uniform probability vector, with marked features from $\mathcal{S}$, denoted as $\mathcal{V}$. This is equal to $O(2^a)$ in the worst-case scenario where a test vector is created for each $a$-feature combination ($i = a$).

Lines 7-13 iterates over each test vector $v \in \mathcal{V}$:

1. Performs a forward pass on the trained autoencoder $AE$ in $O(|\mathcal{F}|)$.

2. Checks minimum probability over the feature subsets in $f \in S$ in comparison to $\tau_a$, in $O(|S|)$.

3. And filters out low support antecedents for the features in $S$, in $O(|S|)$.

Lines 12-13 iterates over the features $\mathcal{F}$ that are not marked, $f \in \mathcal{F} \setminus S$:

1. Checking whether probability $p_f$ exceeds $\tau_c$ in $O(|\mathcal{F}|)$.

2. Stores high-probability features $f$ as consequent and the marked features $S$ as antecedents in $O(|\mathcal{F}|)$ time.

Lastly, line 14 filters low support antecedents in $\mathcal{F}$ in $O(|\mathcal{F}|)$.

**Aggregating** the analysis above results in the following dominant elements:

1. The outer loop runs in $O(a)$ time.

2. The feature subset generation in line 3 runs in $O(|\mathcal{F}|^i)$.

3. Each subset evaluation $S \in \mathcal{C}$ takes $O(|\mathcal{F}|)$ (lines 7-13 in total).

4. Summing over all subset sizes per antecedent-combination from $i = 1$ to $a$:

$$O\left(\sum_{i=1}^{a} |\mathcal{F}|^i \cdot |\mathcal{F}|\right) = O\left(|\mathcal{F}|^{a+1}\right)$$

which leads to $O(a \cdot |\mathcal{F}|^{a+1})$.

Assuming that $a$ is typically a small number, especially for tabular datasets, (e.g., less than 10, and 2-4 for many real-world ARM applications), the final runtime complexity is polynomial over $|\mathcal{F}|$. Following the notation in Section 3 where $|\mathcal{F}| = k$, the runtime complexity of Algorithm 1 in big O is $\boldsymbol{O(k^{a+1})}$ with $a$ being a constant.

Note that the training of the autoencoder is linear over the number of transactions $n$, $O(n)$, as we only perform a forward pass per transaction.

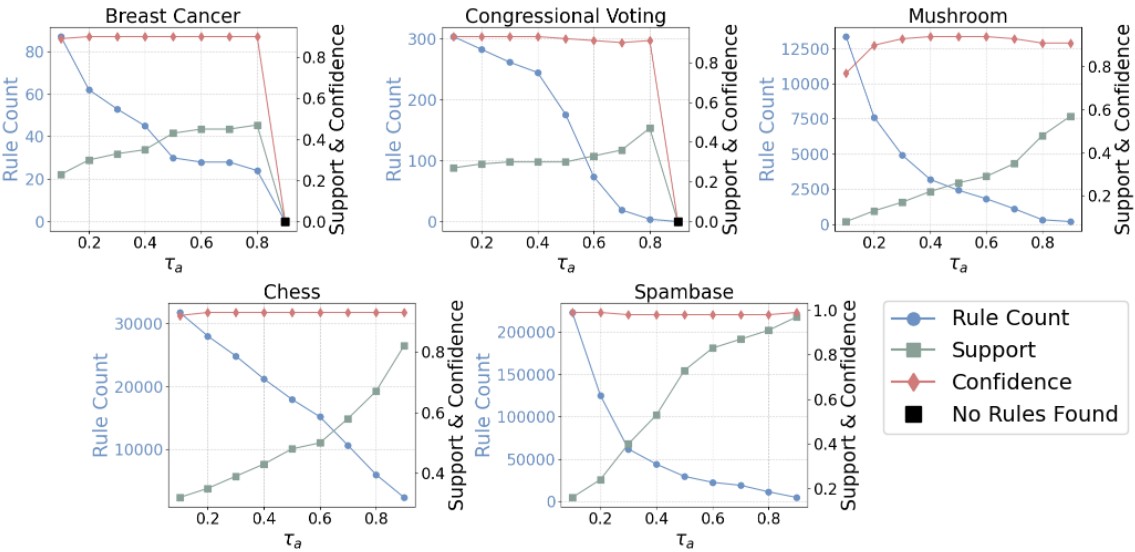

Figure 5: Increasing $\tau_a$ results in a lower number of rules with higher support.

## Appendix B. Hyperparameter Analysis of Aerial+

Aerial+ has 2 hyperparameters besides the number of antecedents which was analyzed in Section 4.1: $\tau_a$ to control antecedent probability threshold and $\tau_c$ to control consequent probability. The section analyzes the effect of $\tau_a$ and $\tau_c$ on rule quality.

**Setup.** We train our autoencoder as described in Section 3.2 and extract rules with 2 antecedents based on varying values of $\tau_a$ and $\tau_c$ on all 5 datasets in 2 sets of experiments. The experiments with varying $\tau_a$ values have $\tau_c$ set to 0.8. The experiments with varying $\tau_c$ values have $\tau_a$ set to 0.5 for Spambase, Chess, and Mushroom datasets, and 0.1 for Breast Cancer and Congressional voting records as the latter are low support datasets.

Figure 5 illustrates the variation in rule count, average support, and confidence values as $\tau_a$ is incremented from 0.1 to 0.9 in steps of 0.1 across all datasets. The findings indicate that an increase in $\tau_a$ leads to a reduction in the number of extracted rules, while the average support of these rules exhibits a consistent upward trend across all datasets. Conversely, the average confidence remains relatively stable, showing minimal variation. Setting the $\tau_a$ to 0.9 did not result in any rules for the Breast Cancer and the Congressional Voting records datasets.

Figure 6 presents the changes in rule count, average support, and confidence values as $\tau_c$ is varied from 0.5 to 0.9 in increments of 0.1 across all datasets. The results demonstrate that as $\tau_c$ increases, both the average support and confidence values exhibit an increasing trend across all datasets, whereas the total number of extracted rules decreases accordingly. Setting the $\tau_c$ to 0.9 did not result in any rules for the Breast Cancer dataset.

## Appendix C. Variations of Aerial+

This section presents two variations to Aerial+'s rule extraction method given in Algorithm 1, and further describes how other ARM variations such as top-k rule mining (Fournier-

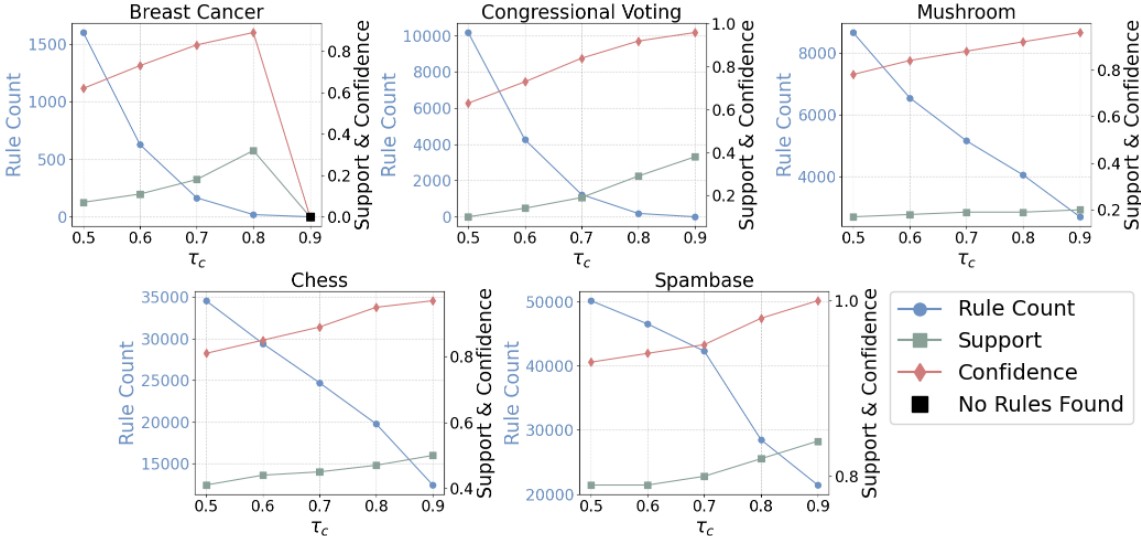

Figure 6: Increasing $\tau_c$ results in a lower number of rules with higher support and confidence.

Viger et al., 2012) can be incorporated into Aerial+. Modifications in the proposed variations relative to Algorithm 1 are distinguished using a light yellow background. Python implementations of the two variants described below (and more) can be found as part of the PyAerial package: https://anonymous.4open.science/r/pyaerial-AF2B.

**Frequent itemset mining with Aerial+.** Algorithm 2 is an Aerial+ variation to mine frequent itemsets. Instead of using antecedent ($\tau_a$) and consequent ($\tau_c$) similarity thresholds, it relies on itemset similarity ($\tau_i$), analogous to $\tau_a$, while eliminating consequent similarity checks. The rationale is that frequently co-occurring items yield high probabilities after a forward pass through the trained autoencoder when those itemsets are marked.

BRL (Letham et al., 2015) and CORELS (Angelino et al., 2018) algorithms require pre-mined frequent itemsets (rather than rules) and class labels to build rule-based classifiers. Algorithm 2 is used to learn frequent itemsets with Aerial+ and the itemsets are then passed to BRL and CORELS to build rule-based classifiers. As described in Section 4.2, frequent itemsets learned by Aerial+ resulted in substantially lower execution times while improving or maintaining classification accuracy, hence, validating the correctness and effectiveness of the proposed Aerial+ variation.

**ARM with item constraints with Aerial+.** Algorithm 3 is an Aerial+ variation for ARM with item constraints. The ARM with item constraints focuses on mining rules for features of interest rather than all features (Srikant et al., 1997). Additional $\mathcal{I}_a$ and $\mathcal{I}_c$ parameters refer to the features of interest on the antecedent side and the consequent side respectively. In line 3, the feature combinations are built using $\mathcal{I}_a$ set rather than all features in $\mathcal{F}$. When checking the consequent similarities between lines 12-14, only the features in $\mathcal{I}_c$ are taken into account. Lastly, the line 15 updates $\mathcal{I}_a$ by removing the low-support features.

CBA (Liu et al., 1998) uses ARM with item constraints to mine rules that have the class label on the consequent side. As part of experimental setting 2 in Section 4.2, Algorithm 3

---

**Algorithm 2:** Frequent itemset mining with Aerial+.

---

**Input:** Trained autoencoder: $AE$, max antecedents: $a$, itemset similarity $\tau_i$

**Output:** Extracted rules $\mathcal{R}$

1   $\mathcal{I} \leftarrow \emptyset$, $\mathcal{F} \leftarrow AE.input\_feature\_categories$;

2   **for** $i \leftarrow 1$ **to** $a$ **do**

3      $\mathcal{C} \leftarrow \binom{\mathcal{F}}{i}$;

4      **foreach** $S \in \mathcal{C}$ **do**

5         $\mathbf{v}_0 \leftarrow \text{UniformProbabilityVectorPerFeature}(\mathcal{F})$;

6         $\mathcal{V} \leftarrow \text{MarkFeatures}(S, \mathbf{v}_0)$

7         **foreach** $\mathbf{v} \in \mathcal{V}$ **do**

8            $\mathbf{p} \leftarrow AE(\mathbf{v})$;

9            **if** $\min\limits_{f \in S} p_f < \tau_i$ **then**

10               $S.\text{low\_support} \leftarrow$ **True**;

11               $\mathcal{I} \leftarrow \mathcal{I} \cup S$;

12      $\mathcal{F} \leftarrow \{f \in \mathcal{F} \mid f.\text{low\_support} = \textbf{False}\}$;

13   **return** $\mathcal{R}$;

---

**Algorithm 3:** ARM with item constraints with Aerial+.

---

**Input:** Trained autoencoder: $AE$, max antecedents: $a$, similarity thresholds $\tau_a, \tau_c$, items of interest $\mathcal{I}_a$ and $\mathcal{I}_c$

**Output:** Extracted rules $\mathcal{R}$

1   $\mathcal{R} \leftarrow \emptyset$, $\mathcal{F} \leftarrow AE.input\_feature\_categories$;

2   **for** $i \leftarrow 1$ **to** $a$ **do**

3      $\mathcal{C} \leftarrow \binom{\mathcal{I}_a}{i}$;

4      **foreach** $S \in \mathcal{C}$ **do**

5         $\mathbf{v}_0 \leftarrow \text{UniformProbabilityVectorPerFeature}(\mathcal{F})$;

6         $\mathcal{V} \leftarrow \text{MarkFeatures}(S, \mathbf{v}_0)$

7         **foreach** $\mathbf{v} \in \mathcal{V}$ **do**

8            $\mathbf{p} \leftarrow AE(\mathbf{v})$;

9            **if** $\min\limits_{f \in S} p_f < \tau_a$ **then**

10               $S.\text{low\_support} \leftarrow$ **True**;

11               **continue** with the next $\mathbf{v}$;

12            **foreach** $f \in (\mathcal{I}_c \setminus S)$ **do**

13               **if** $p_f > \tau_c$ **then**

14                  $\mathcal{R} \leftarrow \mathcal{R} \cup \{(S \rightarrow f)\}$;

15      $\mathcal{I}_a \leftarrow \{f \in \mathcal{I}_a \mid f.\text{low\_support} = \textbf{False}\}$;

16   **return** $\mathcal{R}$;

---

is run to learn rules with class labels on the consequent side, to be able to run CBA, hence validating the correctness and effectiveness of the proposed Aerial+ variation.

We argue that many other ARM variations can be easily incorporated into Aerial+. A third example for this argument is the top-k rule mining (Fournier-Viger et al., 2012). As the experiments in Section B show, higher antecedent and consequent similarity thresholds in Aerial+ result in higher support and confidence rules respectively. In order to mine e.g., top-k rules per consequent, we can simply focus on rules with the top-k highest antecedent support, as part of the checks in lines 9-13 in Algorithm 1.

## Appendix D.  Execution Time Experiments for Optimization-based ARM

The execution time and quality of the rules mined by the optimization-based ARM methods depend on their preset $max\_evaluation$ parameter as presented in Section 4.1. $max\_evaluation$ refers to the maximum number of fitness function evaluations, which are typically a function of rule quality metrics, before termination.

We run the optimization-based methods with the parameters described in Table 2, and with varying numbers of $max\_evaluations$. Table 3 in Section 4.1 presented the results for the Mushroom dataset. Table 6 presents the results for the remaining four datasets.

The results show that on average as the $max\_evaluations$ increases, the number of rules, execution time, and the average confidence of the rules increase while the improvement in the confidence levels stagnates after 50,000 evaluations. Optimization-based methods could not find any rules on the Spambase dataset, except the BAT algorithm, up to 100,000 evaluations which took $\tilde{2}0$ minutes to terminate. The results are consistent with Mushroom dataset results given in Table 3.

## Appendix E.  Effect of Minimum Support Threshold on Classification

This section analyses the effect of minimum support threshold for the exhaustive ARM algorithms on rule-based classification accuracy.

**Setup.** Similar to the experimental setting 2 in Section 4.2, we first run the exhaustive ARM algorithm FP-Growth with different minimum support thresholds and then pass the learned rules (or frequent itemsets for BRL and CORELS) to the three rule-based classifiers. The minimum confidence threshold is set to 0.8 (80%), the number of antecedents is set to 2 and we performed 10-fold cross-validation.

Table 7 shows the change in the number of rules (or itemsets, given under "# Rules") and accuracy based on the preset minimum support threshold on all five datasets.

CBA resulted in higher accuracy with lower support thresholds on all datasets except the Congressional Voting Records. Similar to CBA, the BRL algorithm also led to higher accuracy levels with lower support thresholds on average, with the exception of the Spambase dataset where the accuracy was higher at the 0.3 support threshold. CORELS, on the other hand, had similar accuracy levels for all support thresholds on Congressional Voting Records, higher accuracy with lower support on the Breast Cancer dataset, and did not show a clear pattern on the other three datasets.

Overall, the results indicate that there is no single pattern for selecting rules (whether low or high support) when building a classifier, as it depends on the characteristics of the dataset. Therefore, exhaustive methods often require fine-tuning of the minimum support threshold (or other quality metrics), which can be time-consuming, as mining low-support

Table 6: Optimization-based methods need long evaluations for better performance.

| Evals. | Algorithm | # Rules | Time (s) | Conf. | Evals. | Algorithm | # Rules | Time (s) | Conf. |
|---|---|---|---|---|---|---|---|---|---|
| | | **Chess** | | | | | **Spambase** | | |
| 1000 | BAT | 16.5 | 8.81 | 0.28 | 1000 | BAT | 0 | 14.66 | - |
| | GWO | 12.4 | 12.65 | 0.05 | | GWO | 0 | 19.39 | - |
| | SC | 0 | 12.59 | - | | SC | 0 | 18.6 | - |
| | FSS | 0 | 13.43 | - | | FSS | 0 | 17.77 | - |
| 10000 | BAT | 2241.1 | 52.39 | 0.65 | 10000 | BAT | 2749.2 | 60.08 | 0.56 |
| | GWO | 1278.2 | 73.27 | 0.44 | | GWO | 0 | 93.74 | - |
| | SC | 0.2 | 88.71 | 0.01 | | SC | 0 | 94.78 | - |
| | FSS | 7.8 | 88.82 | 0.29 | | FSS | 0 | 96.23 | - |
| 50000 | BAT | 2905.9 | 235.34 | 0.64 | 50000 | BAT | 10014 | 424 | 0.77 |
| | GWO | 5605.25 | 255.56 | 0.65 | | GWO | 0 | 508 | - |
| | SC | 1 | 545.71 | 0.7 | | SC | 0 | 643 | - |
| | FSS | 32.75 | 380.73 | 0.36 | | FSS | 0 | 677 | - |
| 100000 | BAT | 2816.6 | 529.42 | 0.58 | 100000 | BAT | 9172 | 1316.37 | 0.82 |
| | GWO | 9008.2 | 448.52 | 0.68 | | GWO | 10417.6 | 1704.16 | 0.36 |
| | SC | 0 | 331 | - | | SC | 0 | 1283.59 | - |
| | FSS | 20299.75 | 864.34 | 0.47 | | FSS | 978.2 | 1372.3 | 0.15 |
| | | **Breast Cancer** | | | | **Congressional Voting Records** | | | |
| 1000 | BAT | 144 | 1.3 | 0.32 | 1000 | BAT | 123.9 | 2.34 | 0.36 |
| | GWO | 169 | 1.47 | 0.37 | | GWO | 188.6 | 2.68 | 0.67 |
| | SC | 32 | 1.5 | 0.25 | | SC | 9.2 | 2.53 | 0.33 |
| | FSS | 109 | 1.7 | 0.28 | | FSS | 35.9 | 2.95 | 0.36 |
| 10000 | BAT | 694.2 | 31.64 | 0.38 | 10000 | BAT | 1632.2 | 44.21 | 0.48 |
| | GWO | 707.3 | 31.77 | 0.42 | | GWO | 1018.8 | 40.08 | 0.5 |
| | SC | 29.5 | 39.75 | 0.23 | | SC | 8.1 | 50.04 | 0.31 |
| | FSS | 999.4 | 42.77 | 0.28 | | FSS | 478 | 60.86 | 0.43 |
| 50000 | BAT | 787.1 | 162.18 | 0.41 | 50000 | BAT | 1913 | 208 | 0.45 |
| | GWO | 1584 | 129.18 | 0.42 | | GWO | 2542 | 186 | 0.48 |
| | SC | 33.6 | 137.66 | 0.27 | | SC | 7 | 186 | 0.43 |
| | FSS | 6451.6 | 225.71 | 0.36 | | FSS | 10087 | 272 | 0.71 |
| 100000 | BAT | 750 | 305 | 0.4 | 100000 | BAT | 1856 | 488 | 0.46 |
| | GWO | 2709 | 319 | 0.38 | | GWO | 4035 | 390 | 0.41 |
| | SC | 28 | 310 | 0.25 | | SC | 8 | 421 | 0.67 |
| | FSS | 13523 | 493 | 0.39 | | FSS | 33302 | 992 | 0.85 |

thresholds incurs significant execution time (see execution time experiments in Section 4.1). In Section 4.2, we demonstrate that Aerial+ is more effective at capturing associations between data features that lead to higher accuracy levels more quickly than exhaustive methods.

Table 7: Effect of minimum support threshold for FP-Growth on accuracy when run as part of rule-based classification algorithms on Congressional Voting Records and Spambase datasets.

| Algorithm | Support | # Rules | Accuracy | Algorithm | Support | # Rules | Accuracy |
|---|---|---|---|---|---|---|---|
| **Congressional Voting Records** | | | | **Spambase** | | | |
| CBA | 0.1 | 2200 | 92.22 | CBA | 0.3 | 109 | 84.48 |
| CBA | 0.05 | 2408 | 92.22 | CBA | 0.2 | 163 | 83.78 |
| CBA | 0.01 | 3437 | 91.91 | CBA | 0.1 | 125223 | 84.5 |
| BRL | 0.5 | 25 | 96.97 | BRL | 0.5 | 20389 | 44.16 |
| BRL | 0.3 | 649 | 96.97 | BRL | 0.3 | 24792 | 75.3 |
| BRL | 0.1 | 2547 | 96.97 | BRL | 0.2 | 26224 | 70.46 |
| CORELS | 0.5 | 25 | 96.97 | BRL | 0.1 | 37626 | 72.78 |
| CORELS | 0.4 | 208 | 96.97 | CORELS | 0.3 | 24792 | 87.32 |
| CORELS | 0.3 | 649 | 96.97 | CORELS | 0.2 | 26224 | 84.72 |
| CORELS | 0.1 | 2547 | 96.97 | CORELS | 0.1 | 36737 | 85.15 |
| CORELS | 0.01 | 4553 | 96.97 | CORELS | 0.01 | 275003 | 85.37 |
| **Mushroom** | | | | **Chess (King-Rook vs. King-Pawn)** | | | |
| CBA | 0.3 | 65 | 95.71 | CBA | 0.1 | 32983 | 90.11 |
| CBA | 0.2 | 171 | 99.09 | CBA | 0.05 | 38876 | 90.36 |
| CBA | 0.1 | 5850 | 99.75 | CBA | 0.01 | 49775 | 94.02 |
| CBA | 0.01 | 27800 | 99.82 | BRL | 0.5 | 4046 | 66.11 |
| BRL | 0.5 | 221 | 96.88 | BRL | 0.3 | 8434 | 77.46 |
| BRL | 0.3 | 823 | 99.59 | BRL | 0.2 | 12676 | 94.21 |
| BRL | 0.1 | 5093 | 99.87 | BRL | 0.1 | 19312 | 96.21 |
| CORELS | 0.5 | 221 | 92.91 | CORELS | 0.5 | 4046 | 81.25 |
| CORELS | 0.4 | 413 | 99 | CORELS | 0.4 | 5881 | 94.08 |
| CORELS | 0.3 | 823 | 96.59 | CORELS | 0.3 | 8434 | 90.95 |
| CORELS | 0.2 | 1811 | 94.63 | CORELS | 0.1 | 19312 | 93.46 |
| CORELS | 0.1 | 5093 | 92.46 | CORELS | 0.01 | 37104 | 81.1 |
| CORELS | 0.01 | 23271 | 90.14 | | | | |
| **Breast Cancer** | | | | | | | |
| CBA | 0.1 | 145 | 69.33 | | | | |
| CBA | 0.05 | 273 | 69.32 | | | | |
| CBA | 0.01 | 695 | 66.42 | | | | |
| BRL | 0.1 | 293 | 71.13 | | | | |
| BRL | 0.05 | 655 | 71.5 | | | | |
| BRL | 0.01 | 2047 | 71.13 | | | | |
| CORELS | 0.1 | 293 | 69.73 | | | | |
| CORELS | 0.05 | 655 | 74.37 | | | | |
| CORELS | 0.01 | 2047 | 73.69 | | | | |

## Appendix F. Formal Justification for Aerial+'s Rule Extraction

This section provides a formal justification as to why Aerial+'s rule extraction (Section 3.3) from its Autoencoder architecture (Section 3.2) works in practice.

### F.1. Autoencoder Input

Let $\mathcal{X} \subseteq \{0,1\}^d$ denote the space of binary feature vectors (consisting of one-hot encodings of categorical features). Let $x \in \mathcal{X}$ be a data point composed of $k$ categorical features, i.e., columns of a table, where each feature $x_i$ takes a value in a finite set $\mathcal{A}_i$ of cardinality $C_i$.

Each categorical feature is represented by a one-hot encoding over $C_i$ positions. Thus, the full input vector $x \in \{0,1\}^d$ is a concatenation of such one-hot encodings.

The input is corrupted by adding (or subtracting) uniform noise from the interval $[0,0.5]$ to each entry in $\mathcal{X}$, followed by clipping the result to the range $[0,1]$. The corrupted input is represented with $\tilde{\mathcal{X}}$. Formally, $\tilde{x} \in \tilde{\mathcal{X}}$ is given by:

$$\tilde{x} = \text{clip}(x + \epsilon), \quad \epsilon \sim \mathcal{U}([-0.5, 0.5]^d), \quad \text{clip}(z) = \min(\max(z,0),1). \tag{1}$$

### F.2. Autoencoder Architecture

An under-complete denoising autoencoder consists of:

- **Encoder:** $f : \tilde{\mathcal{X}} \to \mathbb{R}^e$, with $e < d$ (under-complete).

- **Decoder:** $g : \mathbb{R}^e \to \mathbb{R}^d$.

- **Categorical projection layer:** Let $\mathcal{C}_i \subset \{1, \ldots, d\}$ be the index subset corresponding to the one-hot encoding of feature $x_i$. The decoder produces logits $\mathbf{h}_{\mathcal{C}_i} \in \mathbb{R}^{C_i}$, and applies softmax:

$$\bar{x}_{\mathcal{C}_i} = \text{softmax}(\mathbf{h}_{\mathcal{C}_i}) \in \left\{ \mathbf{p} \in \mathbb{R}^{C_i} \;\middle|\; p_j \geq 0, \; \sum_{j=1}^{C_i} p_j = 1 \right\}, \tag{2}$$

where $\bar{x} \in \bar{\mathcal{X}}$ is the reconstructed form of $\mathcal{X}$.

### F.3. Training stage

The model is trained to minimize expected binary cross-entropy loss between the initial input $\mathcal{X}$ and reconstruction $\bar{\mathcal{X}}$:

$$\mathcal{L} = \mathbb{E}\left[ \sum_{i=1}^{n} \text{BCE}(x_{\mathcal{C}_i}, \bar{x}_{\mathcal{C}_i}) \right], \tag{3}$$

Each component $\bar{x}_{\mathcal{C}_i}^{(c)}$, where $c \in \{1, \ldots, C_i\}$, denotes the $c$-th entry of the softmax output vector corresponding to feature $x_i$. Since the model is trained to minimize the binary cross-entropy between the true one-hot vector $x_{\mathcal{C}_i}$ and the predicted distribution $\bar{x}_{\mathcal{C}_i}$, this vector approximates the conditional distribution over categories given the corrupted input:

$$\bar{x}_{\mathcal{C}_i}^{(c)} \approx \mathbb{P}\left( x_{\mathcal{C}_i}^{(c)} = 1 \mid \tilde{x} \right), \tag{4}$$

This approximation is justified by the denoising objective: during training, the model learns to predict the original categorical value from noisy inputs, making the softmax outputs probabilistic estimates of the true class conditioned on the corrupted input.

### F.4. Rule Extraction Stage

To determine whether a specific feature-value assignment implies other feature-value activations, we construct a controlled input probe called a *test vector* and evaluate the decoder's output response.

Let $x_i \in \mathcal{A}_i$ be a categorical feature with domain size $C_i$, and let $c \in \{1, \ldots, C_i\}$ be a particular category. We aim to test whether the assignment $x_i = c$ implies other specific feature values.

We define an artificial input vector $A \in [0, 1]^d$ constructed as follows:

- For the index subset $\mathcal{C}_i \subseteq \{1, \ldots, d\}$ corresponding to the one-hot encoding of $x_i$, set $A_{\mathcal{C}_i}^{(c)} = 1$ and $A_{\mathcal{C}_i \setminus \{c\}} = 0$.

- For all other categorical feature groups $\mathcal{C}_j$ with $j \neq i$, assign uniform probability across the classes: for all $c' \in \{1, \ldots, C_j\}$,

$$A_{\mathcal{C}_j}^{(c')} = \frac{1}{C_j}.$$

This creates an input vector where only $x_i = c$ is deterministically assigned, while all other feature values have uniform probability. No corruption is applied to the test vector.

**Forward pass and output interpretation.** Next, we feed the input $A$ through the trained autoencoder to obtain output:

$$\bar{x} = \mathrm{g}(\mathrm{f}(A)) \in [0, 1]^d,$$

where the decoder output, function $g$, also includes the categorical projection layer step. For each feature $j$, $\bar{x}_{\mathcal{C}_j} \in \mathbb{R}^{C_j}$ is a probability vector over its possible categories, derived from the softmax output of the decoder.

**Antecedent similarity confirmation.** Confirm that the decoded output preserves the input's forced condition by checking:

$$\bar{x}_{\mathcal{C}_i}^{(c)} \geq \mathcal{T}_a,$$

where $\mathcal{T}_a \in (0, 1)$ is an antecedent similarity threshold. This ensures that the model recognizes $x_i = c$ as a highly probable input condition.

**Consequent discovery.** Search the output $\bar{x}$ for all components $\bar{x}_{\mathcal{C}_j}^{(c')}$ such that:

$$\bar{x}_{\mathcal{C}_j}^{(c')} \geq \mathcal{T}_c, \quad \text{for } j \neq i \text{ (no self-implication)},$$

where $\mathcal{T}_c \in (0, 1)$ is a consequent similarity threshold. The set of all such $(x_j = c')$ pairs forms the set of predicted consequents:

$$B = \left\{ (x_j = c') \,\middle|\, \bar{x}_{\mathcal{C}_j}^{(c')} \geq \mathcal{T}_c, \ j \neq i \right\}.$$

If both the antecedent similarity and the consequent similarity conditions are satisfied, we conclude with the following rule:

$$(x_i = c) \Rightarrow \bigwedge_{(x_j = c') \in B} (x_j = c'). \tag{5}$$

Note that, the same can be applied for multiple features $x_i$ on the antecedent side. Since the decoder output approximates conditional probabilities learned during denoising training,

$$\bar{x}_{\mathcal{C}_j}^{(c')} \approx \mathbb{P}(x_j = c' \mid x_i = c), \tag{6}$$

this probing reveals high-confidence associations in the learned conditional distribution.

### F.5. Interpretation of the formal justification

Overall interpretation of the formal justification can be summarized as follows:

- **Under-completeness.** The encoder maps the corrupted higher-dimensional input $\tilde{x}$ to a lower-dimensional latent space, encouraging the model to capture prominent co-occurrence (association) patterns.

- **Reconstruction objective.** Given the noise process in Equation 1, the model is trained to reconstruct the original input $x$, approximating:

$$\bar{x}_{\mathcal{C}_i}^{(c)} \approx \mathbb{P}(x_i = c \mid \tilde{x}).$$

- **Categorical output.** The decoder outputs per-feature softmax distributions over categories. Thus, $\bar{x}_{\mathcal{C}_i}^{(c)} \in [0, 1]$ quantifies the model's belief that feature $x_i$ takes value $c$, conditioned on the input.

- **Probing with test vectors.** Rule extraction relies on probing the decoder with a constructed input $A$ that sets $x_i = c$ deterministically and others to uniform noise. The resulting output $\bar{x}$ estimates $\mathbb{P}(x_j = c' \mid x_i = c)$ (Equation 6), enabling the discovery of association rules.

**Conclusion.** A denoising autoencoder trained on the corrupted categorical data learns a conditional distribution over feature values. The decoder's categorical projection layer approximates $\mathbb{P}(x_j = c' \mid x_i = c)$, making them suitable for rule extraction via direct probing. This forms the basis of Aerial+'s neurosymbolic association rule mining.

