# OpenReview forum: "Neurosymbolic Association Rule Mining from Tabular Data"
_nesyconf.org/NeSy/2025/Conference_Phase_2 — NeSy 2025 - Phase 2 Poster_

### Official Review · Reviewer_oQzU · 2025-06-23
**Good answers to reviewer's questions but still not a NeSy paper**

**Rating:** 4
**Confidence:** 4

**Review:**

The authors answered most of my questions. However, I am still convinced that this is not a NeSy paper because it does not leverage any symbolic knowledge. It just produces a symbolic output. It is a good Deep Learning paper for a pure Machine Learning venue that solves in an efficient way a combinatorics problem.

**Anonymity:**

Disclose identity

---

### Official Review · Reviewer_7GYT · 2025-07-07
**Good attempt but scalability issues need to be addressed properly**

**Rating:** 6
**Confidence:** 4

**Review:**

Regarding the scalability, a and  |F| are not constants as they vary from problems to problems. The complexity, as the authors mentioned, is O(|F|^a) which is exponential.  There is no formal proof for the bounds of the max antecedents, therefore it does not guarantee that the maximum number of antecedents is always small when  scaling up. Currently, this is still a hyper-parameter. Let’s take an example if  there are 1000 features with 10 possible antecedents, the size of set C will be 2.634e+23, this is a huge number. Note that, in the experiment, the highest number of features is just 57 (Spambase), this is not always the case in real-life applications.

The search for this hyper-parameter a will also a question.

Please note that the Big O notation in Line 3 Appendix A is not correctly presented.

Before GAN and VAE, AutoEncoder and Restricted Boltzmann Machines were considered “generative models” to distinguish with “discriminative models” where inputs and outputs are treated differently, see Pascal et. al ICML 2008. The question about “structure” of AE, meaning whether the extracted rules can help interpret the AE or not.

Rule learning is also for association rules,  Column Generation, Bayesian Rule Sets (BRS), RIPPER, CART, DR-Net, RL-Net, etc.

I appreciate the attempt of the authors in improving the paper. Although the scalability issues still persist, there can be an extensions from this study to deal with it in a future work.

**Anonymity:**

Remain anonymous

---

### Official Review · Reviewer_qTLy · 2025-07-09
**Improved presentation, useful method**

**Rating:** 7
**Confidence:** 3

**Review:**

The new version of the paper has improved presentation. The results are essentially the same (I haven't had time to check in detail).
I feel that this is a useful method and with the clarification good for acceptance. The argument for a limited number of antecedent makes the complexity indeed polynomial, but that is not necessarily so for unlimited antecedents combinatorial explosion applies.

Small issues:
Section 2 ARM presentation: The support should be defined as a ratio, not in percent s%
Section 3.2 Learning Rate 5e^{−3} looks like Euler's number, should use scientific (5E-3) or mathemtical (5 * 10^{-3}) notation. The same goes for weight decay.

**Anonymity:**

Remain anonymous